# Vegetation change over seven years in the largest protected Pacific Northwest Bunchgrass Prairie remnant

Joshua P. Averett[1]*, Lesley R. Morris[1], Bridgett J. Naylor[2], Robert V. Taylor[3], Bryan A. Endress[1]

**1** Eastern Oregon Agricultural Research Center, Oregon State University, Union, Oregon, United States of America, **2** USDA Forest Service, Pacific Northwest Research Station, La Grande, Oregon, United States of America, **3** National Wildlife Refuge Association, Enterprise, Oregon, United States of America

* averettj@oregonstate.edu

**Data Availability Statement:** The data article and all supplementary files are available (In Press, Pre-proof) at: https://doi.org/10.1016/j.dib.2019.105104.

## Abstract

Temperate grasslands are one of the most altered ecosystems on Earth. Consequently, conservation of important characteristics of such ecosystems (e.g., biodiversity) is uncertain even within grasslands that have been protected. Invasion by non-native plants is considered a primary threat to intact grasslands. Here, we evaluated native and non-native vegetation composition change over seven years in the largest Pacific Northwest Bunchgrass remnant. We sampled 124 permanent plots across the Zumwalt Prairie Preserve (northeastern Oregon, USA) twice, seven years apart. With data collected from three grassland community types (xeric prairie, mesic prairie, old fields), we asked: (1) how has species composition changed over time; (2) which species showed the greatest changes in abundance; and (3) how did abundance of *Ventenata dubia* (the most abundant non-native species) relate to patterns of native and non-native plant abundance? Vegetation composition changed in all three plant communities. *Ventenata dubia*, an annual non-native grass: (1) became the third most dominant species across the study area; (2) was the only non-native that increased in abundance substantially in all three communities; and (3) was negatively related to native perennial forb cover. Relative cover of non-native species decreased in old fields concomitant with increases in native bunchgrass (*Festuca idahoensis*) and *V. dubia* cover. Increased cover of native perennial grasses and non-native annual grasses in old fields were associated with loss of bare ground, but not with reductions in non-native perennial grass cover. Native species dominated in the mesic prairie; however, non-native cover (particularly *V. dubia*) increased (mean cover increased from 3 to 10%) while mean native perennial forb cover decreased (from 30 to 25%) over time. Continued shifts towards non-native annual grass dominance coupled with potentially declining native perennial forbs, may challenge conservation efforts in one of the last large tracts of Pacific Northwest Bunchgrass Prairie.

**Funding:** Funding was provided by USDA NRI Competitive Grant Agreement No. 2006-35320-17244 (BE), USDA Forest Service Pacific Northwest Research Station # 05-JV-11261967-069 (BE), the Oregon State University Agricultural Research Foundation (LM, BE), and the Oregon State University Agriculture Experiment Station (LM, BE). The funders had no role in study design, data collection and analysis, decision to publish, or preparation of the manuscript.

**Competing interests:** The authors have declared that no competing interests exist.

## Introduction

Temperate grasslands are among the most threatened biomes on Earth and are the most altered ecosystems in North America [1]. Only about 1% of North America's temperate grasslands are in a relatively natural state, and even those that remain have been subjected to significant anthropogenic stressors, e.g., widespread livestock grazing, intentional and accidental introduction of non-native species, and localized settlement/cultivation and subsequent abandonment [1–2]. Because of widespread alteration to temperate grasslands in North America, the ability of these systems to maintain the ecological services, such as biodiversity, that they have historically provided is uncertain [1,3]. Invasion by non-native plant species has been identified as one of the most serious threats to ecosystem function and native biodiversity in remaining temperate grassland areas [4]. Non-native plant invasions have altered grassland structure, reduced floral and faunal diversity, altered disturbance regimes, and, in some circumstances have resulted in complete shifts in species composition and ecosystem function [5–6]. Although plant invasions, to date, have not resulted in species extinctions [7], they may ultimately lead to the loss of species over time [8]. Identification of invading non-natives plants and the dynamics of their spread including associations with native vegetation can provide critical information for the prioritization of conservation, research and monitoring efforts, evaluation of the potential to control and/or adapt to invasions, and understanding the potential impacts of invasion on native communities.

The Zumwalt Prairie Preserve (hereafter ZPP), the focus of this study, encompasses around 13,350 ha in northeastern Oregon, USA, and is located within the largest remaining intact Pacific Northwest Bunchgrass Prairie (the larger Zumwalt Prairie, ~ 65,000 ha). The Pacific Northwest Bunchgrass Prairie (hereafter PNB), once occupied ~ 8 million ha of Oregon, Washington, Idaho and portions of Western Canada [9–11]. It is also one of the least described and least studied of the major vegetation types in North America despite the endangered status for portions of the PNB [9,12]. Consequently, the ZPP was purchased by The Nature Conservancy (hereafter TNC) in two land acquisitions (~ 11,000 ha in 2000 and ~ 2,400 ha in 2006) and is managed to support the conservation of habitat for native species including the largest known population of the threatened Spalding's catchfly (*Silene spaldingii*; [13]) as well as diverse native forb (~64% of vascular plant species [14, 15]) and bee (more than 200 species in 27 genera [15]) communities. Though protected, the ZPP is subject to anthropogenic stressors that are considered threats to conservation of native temperate grasslands (e.g., legacy effects of past cultivation, livestock grazing, invasion by non-native plants; [1,4]). Past research has evaluated the effects of cattle (*Bos taurus*) grazing on vegetation structure, and bird and bee habitat requirements [15–16], the potential to restore old fields that had been abandoned following cultivation [17–18], and impacts of non-native plants on passerine birds [19].

Invasion by non-native plant species, although a recognized threat to the PNB, has been studied less than other disturbances on the PNB. Two studies have documented an increase in *Ventenata dubia* in PNB canyon grasslands since the 1980s and dominance by the invasive annual grass in soil seed banks of old fields [20–21]. A previous evaluation (2008 and 2009) of non-native plant distributions conducted within the upland prairie portion (~ 5,100 ha)–excluding canyons–of the ZPP revealed that non-native grass species were widespread and among the most dominant vascular plant species across the ZPP by 2008 [22]. The most abundant non-native species were *V. dubia*, *Poa pratensis* and annual *Bromus* species (*Bromus* spp; comprised of *Bromus arvensis* and *Bromus hordeaceous*) which had the fourth, fifth, and seventh highest total foliar cover respectively of all species across the study area [22]. Endress et al. ([22]) described three broad plant communities within ZPP where non-native composition varied substantially. The communities identified (xeric prairie, mesic prairie, old fields)

aligned well with other bunchgrass plant associations developed for the region that co-vary primarily with gradients in soil depth and water availability [9,23–24]. Non-native annual grasses (*Bromus* spp. and *V. dubia*) were most abundant in the xeric prairie (highest heat load and greatest soil moisture deficit) associated with increased dominance of the native bunchgrasses *Poa secunda*, *Pseudoroegneria spicata* and *Danthonia unispicata*, and were least abundant in the wettest/deepest soil sites (mesic prairie, and old field communities; [22]). This pattern of apparently low invasion resistance of the more xeric sites to non-native annual grass invasion was consistent with previous observations [23–24] in the region. Non-native perennial grasses (e.g., *P. pratensis* and *Thinopyrum intermedium*) were dominant in or near sites that had been previously cultivated (e.g., old fields). The detection of less invaded mesic communities coexisting alongside annual grass invaded xeric prairie and non-native perennial grass dominated old fields may suggest that never-plowed mesic prairie communities are more resistant to invasion or, alternatively, that invasion by non-native grasses from either the drier or wetter communities is inevitable [22]. The differences in non-native plant composition and abundance across community types in the ZPP suggest that some communities may be more susceptible to invasion and alteration of species composition over time, while other communities may be more resistant to change. There are currently no studies that address vegetation change at a large scale over time in this system.

Our objectives were to analyze vegetation composition change over seven years across the ZPP. For this, we resampled (in 2015 and 2016 –seven years later) permanent plots that were established and initially sampled in 2008 and 2009 within the three community types described by Endress et al. ([22]). We addressed the following questions: (1) how did species composition change over time in the three plant communities; (2) which species showed the greatest changes in abundance; and (3) how did abundance of *V. dubia* (most abundant non-native species) relate to the abundance of native perennial forbs, native perennial grasses, non-native perennial grasses, and other non-native annual grasses?

## Methods

Field sampling took place on private lands within the ZPP. Permission for sampling was granted by and in collaboration with the landowner, The Nature Conservancy. Field sampling for this study was non-destructive, and did not harm endangered or protected species.

### Study site

The ZPP is located in northeastern Oregon (45.577745 N 116.971754 W). Elevations range from 1060 to 1680 m. Precipitation averages ~490 mm annually, mostly occurring in winter and spring, with a pronounced summer drought. Average daily temperatures range from -3 ˚C in winter to 16 ˚C during the summer [10,17]. Seasonal cattle grazing has been the dominant land use on the ZPP for at least five decades [10]. At the turn of the 20th Century, the entire PNB including the ZPP was homesteaded and used for cultivation and then largely abandoned and consolidated into larger parcels under private ownership [10]. Prescribed fire and targeted herbicide spraying (spot spraying of individual plants) for control of some non-native perennial forb species (e.g., localized populations of *Hieracium caespitosum*, *Potentilla recta*, *Onopordum acanthium*) are other important land management practices.

Endress et al. ([22]) used cluster analysis and non-metric multidimensional scaling to describe three plant communities in the ZPP as well as to relate patterns of native and non-native species abundance across those communities to each other and important abiotic/environmental correlates. This study used the same community classification developed by Endress et al. ([22]): (1) the "xeric prairie" community consisted of the most water limited sites

(highest heat load and greatest soil moisture deficits) and was dominated by *V. dubia*, *Poa secunda*, *Pseudoroegneria spicata* and *Danthonia unispicata*. This community aligned well with both the *Poa secunda/Danthonia unispicata* (i.e., shallow soil "scablands") and *Pseudoroegneria spicata/Poa secunda* plant association series described by Johnson and Simon (1987 [23]) and Johnson and Swanson (2005 [24]); (2) The "mesic prairie" community was dominated by *Festuca idahoensis*, *Koeleria macrantha*, *Pseudoroegneria spicata*, and numerous perennial forbs including *Geum triflorum* and *Arnica sororia*. This community also had the least non-native cover in 2008/2009, and aligned mostly with the *Festuca idahoensis/Koeleria macrantha* and *Festuca Idahoensis/Pseudoroegneria spicata* plant association series in the region [23–24]; and (3) the "old field" community consisted of flat or gently sloping sites with deep soils dominated by non-native perennial forage grasses including *Poa pratensis* and *Thinopyrum intermedium*. This community was strongly associated with past cultivation, i.e. old fields and did not align well with previous native plant community descriptions developed in the region; however, the most abundant native species (*Festuca idahoensis*, *Pontentilla gracilis*, *Geum triflorum*) in the old fields were also among those dominant in the mesic prairie.

Non-native grass species are also among the most abundant and widespread species at the ZPP, but generally have not been used to define community variation, e.g., plant association classification (but see [20, 22]). In 2008/2009, non-native annual grasses (*Ventenata dubia and Bromus spp*.) were strongly associated with the most water-limited sites. Both of these annual grasses were widespread across the xeric and mesic prairie; however *Ventenata dubia* cover declined abruptly in the more mesic sites particularly within the old fields [22]. *Bromus* spp. were more uniformly distributed across the xeric and mesic prairie sites compared to *Ventenata dubia*, but were similarly sparse in old fields [22]. Non-native perennial grasses including *Poa pratensis* and *Thinopyrum intermedium* dominated in relatively flat, mesic sites that were associated with old fields established long before the establishment of the preserve [22].

## Field data

In order to evaluate vegetation change over the seven years, vascular plant composition of the most dominant species (based on foliar cover observed in 2008/2009; [22]) were re-sampled within 124 permanent plots (1 ha/each) across the prairie habitat (excluding canyonlands) of the ZPP. For the repeated sample, we focused on the most common species for our analyses (Table 1) in order to reduce observer error associated with measuring rare species, and to increase the number of plots that could be sampled. Further support for our decision to focus on the dominant species was provided by ordination analyses which showed that inclusion of more species did not substantially change ordination structure or findings from the initial (2008/2009) survey. Pilot sampling and review of TNC vegetation monitoring data [17] occurred prior to the resampling effort to ensure that the 31 dominant species selected from the initial sampling effort were still the most abundant species in our study area and that other species had not increased enough to warrant inclusion. Any other vascular plant species encountered during sampling were recorded into functional groups (native perennial grass (NPG), native perennial forb (NPF), native annual grass (NAG), native annual forb (NAF), native sedge/rush, non-native perennial grass (NNPG), non-native perennial forb (NNPF), non-native annual grass (NNAG), non-native annual forb (NNAF)). Vegetation sampling took place during two different time periods, seven years apart: (1) summers of 2008 and 2009; and (2) summers of 2015 and 2016. Plot sampling within each time period was staggered one year so that the plots that were sampled in 2008 were resampled in 2015 and plots that were sampled in 2009 were measured in 2016 resulting in a seven year time gap between resamples for each plot.

**Table 1. Species abundance summaries for dominant vascular plants occurring in the Zumwalt Prairie Preserve study area ordered top to bottom by rank abundance (% cover) from initial time period (T1, 2008/2009).** Log abundance (Log Abund) was calculated as the log (base 10) of the total summed abundance for a particular species in all plots. T2 is the second time period (2015/2016). Non-native species are listed in bold text.

| Species | Rank Abund | | Log Abund | | Rank Freq | | Freq | | Mean Cover (%) | |
|---|---|---|---|---|---|---|---|---|---|---|
| | T1 | T2 | T1 | T2 | T1 | T2 | T1 | T2 | T1 | T2 |
| *Festuca idahoensis* | 1 | 1 | 3.47 | 3.49 | 1 | 1 | 122 | 123 | 24.0 | 24.8 |
| *Psuedoroegneria spicata* | 2 | 2 | 3.30 | 3.30 | 4 | 4 | 107 | 107 | 16.3 | 16.1 |
| *Poa secunda* | 3 | 4 | 3.00 | 3.05 | 2 | 2 | 116 | 112 | 8.1 | 9.1 |
| **Ventenata dubia** | **4** | **3** | **2.97** | **3.27** | **11** | **3** | **84** | **108** | **7.6** | **14.9** |
| **Poa pratensis** | **5** | **6** | **2.87** | **2.85** | **12** | **13** | **75** | **72** | **6.0** | **5.8** |
| *Geum triflorum* | 6 | 5 | 2.86 | 2.95 | 10 | 9 | 87 | 91 | 5.9 | 7.2 |
| **Bromus spp**. | **7** | **7** | **2.79** | **2.67** | **8** | **8** | **95** | **93** | **5.0** | **3.8** |
| *Arnica sororia* | 8 | 9 | 2.71 | 2.58 | 5 | 7 | 104 | 93 | 4.1 | 3.1 |
| **Thinopyrum intermedium** | **9** | **8** | **2.64** | **2.64** | **23** | **18** | **28** | **41** | **3.5** | **3.5** |
| *Lupinus spp.* | 10 | 11 | 2.64 | 2.49 | 3 | 6 | 108 | 94 | 3.5 | 2.5 |
| *Potentilla gracilis* | 11 | 13 | 2.58 | 2.40 | 9 | 12 | 91 | 79 | 3.1 | 2.0 |
| *Achillea millefolium* | 12 | 10 | 2.56 | 2.52 | 6 | 5 | 101 | 99 | 2.9 | 2.7 |
| *Koeleria macrantha* | 13 | 14 | 2.40 | 2.38 | 7 | 10 | 98 | 85 | 2.0 | 1.9 |
| *Bromus carinatus* | 14 | 12 | 2.26 | 2.42 | 18 | 14 | 47 | 64 | 1.5 | 2.1 |
| *Danthonia unispicata* | 15 | 16 | 2.18 | 2.34 | 14 | 16 | 52 | 48 | 1.2 | 1.7 |
| *Danthonia intermedia* | 16 | 22 | 2.15 | 2.02 | 19 | 24 | 46 | 30 | 1.1 | 0.8 |
| **Phleum pratense** | **17** | **19** | **2.10** | **2.15** | **27** | **26** | **22** | **22** | **1.0** | **1.2** |
| **Poa compressa** | **18** | **23** | **2.03** | **1.88** | **24** | **22** | **26** | **35** | **0.9** | **0.6** |
| *Achnatherum occidentale* | 19 | 17 | 2.00 | 2.29 | 25 | 25 | 25 | 23 | 0.8 | 1.6 |
| **Veronica arvensis** | **20** | **15** | **1.99** | **2.36** | **17** | **11** | **47** | **81** | **0.8** | **1.8** |
| *Eriogonum spp.* | 21 | 20 | 1.97 | 2.11 | 13 | 15 | 54 | 52 | 0.7 | 1.0 |
| *Balsamorhiza incana* | 22 | 25 | 1.95 | 1.74 | 20 | 20 | 42 | 36 | 0.7 | 0.4 |
| **Bromus inermis** | **23** | **24** | **1.94** | **1.79** | **28** | **28** | **10** | **12** | **0.7** | **0.5** |
| *Geranium viscosissimum* | 24 | 27 | 1.88 | 1.64 | 22 | 27 | 32 | 18 | 0.6 | 0.4 |
| *Phlox spp.* | 25 | 26 | 1.88 | 1.68 | 15 | 21 | 50 | 35 | 0.6 | 0.4 |
| **Bromus tectorum** | **26** | **18** | **1.86** | **2.24** | **26** | **17** | **24** | **46** | **0.6** | **1.4** |
| *Hieracium cynoglossoides* | 27 | 29 | 1.79 | 1.59 | 16 | 23 | 48 | 31 | 0.5 | 0.3 |
| **Poa bulbosa** | **28** | **30** | **1.73** | **1.08** | **31** | **31** | **3** | **4** | **0.4** | **0.1** |
| *Epilobium brachycarpum* | 29 | 28 | 1.59 | 1.62 | 21 | 19 | 41 | 40 | 0.3 | 0.3 |
| **Festuca rubra** | **30** | **21** | **1.55** | **2.05** | **30** | **30** | **5** | **7** | **0.3** | **0.9** |
| *Solidago canadensis* | 31 | 31 | 1.00 | 0.90 | 29 | 29 | 8 | 10 | 0.1 | 0.1 |

Cattle were grazed at a stocking rate of 4.7 acres per AUM TNC, personal communication, Preserve manager) over most of the study area (95% of plots) for the duration of our study. This stocking rate is consistent with grazing management in the larger Zumwalt Prairie and historical grazing (at least over the last five decades) on the ZPP [10]. Twelve plots (~ 10%) were exposed to some level of management/restoration activities (e.g., prescribed fire at least two years prior to sampling, n = 5; recent or temporary exclusion of cattle grazing, n = 6, herbicide and seeding, n = 1) over the duration of our study [25]. The small sample size of plots with any specific management treatment did not allow for the effects of such treatments to be evaluated in this study. Analyses revealed that comparisons between datasets with deletion and inclusion of the 12 plots subjected to management/restoration treatments did not substantially alter our results. Specifically, no trends or significance of comparisons/relationships were

altered due to the inclusion or deletion of these plots. Therefore, we included all of the plots in our analyses to capture more variation across the prairie habitat of the ZPP.

Within each plot, three line-point intercept transects (length = 50 m) were established in a spoke design, radiating out from the center of the plot at 0˚, 120˚, and 240˚ relative to magnetic North [22,26]. We observed species intercepts with transects at 1 m increments along transects for a total of 150 points sampled (50 per transect) per plot. Because multiple species are often encountered within different canopy layers at the same point, total cover can be > 100%. We calculated foliar cover for each species in a plot as the total number of hits for that species divided by 150 points multiplied by 100. Species frequency for a particular species was the total count of plots occupied by that species (maximum frequency possible = 124). At each sample point, we recorded presence of litter and soil surface characteristics (rock, biotic crust, bare ground) to allow for calculations of cover for each metric. Plot cover of bare ground was calculated as the percentage of hits (out of 150) for bare ground in that plot where no other plant species or soil cover (litter, rock, moss/lichen) were encountered. Sampling occurred within three plant community types (xeric prairie, n = 31; mesic prairie, n = 77; old fields, n = 16; [22]).

## Data analysis

**Vegetation change over time.**   Paired t-tests were used to test the null hypothesis of no-change between non-native relative cover (proportional cover of non-native species relative to total plant cover expressed as a percentage) between sampling periods (plots were the sample units) for our entire study area and for each community type separately. Foliar cover change for functional groups (NPG, NPF, NNPG, NNAG) was calculated and compared to the null hypothesis of no change over time. Evidence of change in foliar cover was evaluated for all plots together and then separately for each community type. Bootstrapping was used (10,000 replicates) to generate 90 and 95% confidence intervals (basic bootstrap interval; [27]) for the mean change (mean of the paired differences; plots were sample units) in foliar cover between sample periods [28–29]. Strong evidence of change in foliar cover was indicated when 95% CIs did not cross zero; weak evidence for change was indicated when 90% CIs did not cross zero.

Blocked Multi Response Permutation Procedure (MRBP; [30–31]) was used to test the null hypothesis of no change in vascular plant species composition between time periods. MRBP is a nonparametric method for testing of no difference between two or more groups after accounting for block differences and provides an effect size chance-corrected within group agreement (*A*-statistic) that indicates within group homogeneity compared to random expectation [32]. An *A*-statistic value = 1.0 indicates that all components of a group are identical, while *A* = 0 means within group homogeneity equals random expectation [32]. We used a Euclidean distance measure and aligned the medians to zero for all blocks (plots were the blocking factor). We performed separate tests for each community type.

Non-metric multidimensional (NMS; [31]) scaling using a Sorensen distance measure was used to extract the dominant community composition gradients from our dataset. We used the "slow and thorough" NMS autopilot setting and Kruskal's strategy 1 for penalization of ties in the distance matrix. A Sorensen distance measure was used to focus on relative differences in species abundances between sample units. Two dimensional ordinations were produced with a random starting configuration and a maximum of 500 iterations. We evaluated linear relationships between community composition variation and functional group abundance using a joint plot and identified community types on the ordination using convex hulls. The community types were defined by the 2008/2009 dataset (see [22] for methods). Temporal vectors that connect each plot between time periods were overlain onto the ordination and translated to the origin. This resulted in the tail of the vector (plot at the initial time of sampling)

being located at the origin and the strength and direction of composition change for a plot being represented by the length and direction of the resulting vector respectively [32].

**Species that changed the most in each community.** Indicator species analysis (ISA) was performed for each of the three community types separately to identify time period tendencies for species, i.e., to identify the species most responsible for driving compositional change between sampling periods [32–33]. ISA produces indicator values (IV) based on relative frequency (proportion of sample units from each group occupied) and abundance (proportional abundance for a particular species in a group relative to abundance in all groups) information that represent the strength of tendencies for species to occur within a priori groups [32]. Therefore, a species with a significant IV for time period one indicated higher abundance for that species during time period one (2008 & 2009) compared to time period two (2015 & 2016). Higher IV values indicate stronger affinity to a particular group (time period in this case). Zero means a species is not an indicator; and 100 represents perfect indication, i.e., the species is always and only observed in that particular group [32]. Significance of IV's were determined by comparing observed values to results from 10,000 randomizations [31–32]. Changes in relative abundance and relative frequency over time were evaluated for identified indicator species. Additional 2-dimensional NMS ordinations were generated for the old field and mesic prairie communities (separately) to evaluate linear relationships between functional group change over time and community variation.

***Ventenata dubia* relationships with native/non-native plant abundance.** Quantile regression was used to explore relationships between the foliar cover of the most abundant non-native species on ZPP, *V. dubia*, and the foliar cover of: (1) native perennials (forbs and grasses combined); (2) native perennial forbs; (3) native perennial grasses; (4) non-native perennial grasses; and (5) other non-native annual grasses separately [28,34]. Quantile regression is a method for estimating relationships between variables for all portions of the probability distribution, is non-parametric, and unlike ordinary least squares regression, does not require homogeneity of variance–an assumption not met in our dataset [35]. We evaluated relationships between the response and predictor variables for the 0.5 and 0.9 quantiles. This allowed us to evaluate functional relationships between variables at the median of the response distribution as well as near the maximum of the response distribution. We evaluated the 0.9 quantile because when there are many unmeasured factors that are potentially limiting, a common occurrence in natural systems, quantiles near the maximum response often provide more meaningful information regarding relationships between variables [35].

## Results

### Vegetation change over time

Non-native relative cover increased across our study area from 27 to 30% over time (two sample paired t-test, two-tailed, t = 2.6, p-value = 0.01; Table 2). This change was primarily focused

**Table 2. Non-native relative cover including difference (Time period 2—Time period 1) and 95% confidence intervals for all plots combined (all) and each vegetation group defined from cluster analysis of species composition for time period 1(2008/2009).**

| Vegetation Group | Non-native Relative Cover (%) | | Difference (CIs) |
|---|---|---|---|
| | 2008/2009 | 2015/2016 | |
| All | 27.10 | 30.23 | **3.13 (0.72, 5.73)** |
| Old Fields | 68.08 | 60.98 | -7.09 (-15.51, 2.98) |
| Mesic Prairie | 16.23 | 22.33 | **6.11 (3.40, 8.46)** |
| Xeric Prairie | 32.98 | 33.98 | 1.00 (-3.17, 5.70) |

within the mesic prairie, where non-native relative cover increased from 16 to 22% (paired t = 4.6, p < 0.001; Table 2). There was weak evidence for a decrease (from 68 to 61%; paired t = -1.5, *p* < 0.14) in non-native relative cover in the old field community, and no evidence of change in the xeric prairie (paired t = 0.45, *p* = 0.66; Table 2). The two most dominant species in terms of cover (*F. idahoensis* and *P. spicata*) and frequency (*F. idahoensis* and *P. secunda*) were consistent over time. Other natives including *Lupinus* spp. (particularly in terms of frequency) and *Danthonia intermedia* (cover and frequency) showed decreased dominance over time (Table 1). Cover and frequency of *V. dubia* increased substantially across our study area; *V. dubia* moved up a rank from fourth to third when evaluated based on cover (mean cover almost doubled over time from 7.6 to 14.9%, Table 1) and from eleventh to third for frequency, making it the third most abundant species in terms of cover and frequency in our study area (Table 1). Rank abundances for annual *Bromus* spp. and *P. pratensis*–the second and third most abundant non-natives- were relatively consistent over time. *T. intermedium* (frequency increased by 10%), and the annual forb *Veronica arvensis* (mean cover more than doubled from 0.8 to 1.8% and frequency increased by 27%) were two other non-native species that showed signs of increased dominance; however their mean cover and frequency were much lower than *V. dubia* (Table 1; [25]).

Non-native annual grass cover increased from 13 to 20%, and native perennial forb cover decreased from 23 to 20% over time (Fig 1). There was weak evidence that native perennial grass cover increased from 55 to 58% (change = 3.1% 90% CI = 0.6, 6.0) since the initial sampling effort (Fig 1). Increases in non-native annual grass mean cover appeared to be focused primarily in the old field (increased from 4 to 16%) and mesic prairie communities (increased from 8 to 16%; Fig 1). Mean native perennial forb cover decreased (from 30 to 25%) in the mesic prairie (Fig 1). Mean native perennial grass cover increased substantially (from 12 to 27%) in old fields (Fig 1), and was consistent with an increase in total vascular plant cover (11.6%, 95% CI = 2.4, 20.7). There was no evidence for change in cover for any functional group within the xeric prairie (Fig 1). We also calculated the total change in native perennial forb cover (not limited to the dominant species) to make sure that trends in dominant forbs tracked the entire community. Consistent with the dominant forbs, all native perennial forbs decreased by 5% (95% CI = -7.6, -2.9) in our sample area, and the greatest decrease for all native perennial forb foliar cover (-6.6, 95% CI = -9.9, -3.2) was observed in the mesic prairie.

Vegetation composition changed over time in all three plant communities (p < 0.001; Table 3). The greatest changes were detected in old fields (*A* = 0.07) followed by the mesic prairie (*A* = 0.05), and lastly by the xeric prairie (*A* = 0.03; Table 3). Temporal vectors extracted from the NMS ordination indicated that most old field plots shifted to lower Axis 1 (explaining 64% of variation in the distance matrix) scores over time which was associated with lower composition of non-native perennial grass species (Fig 2). The mesic prairie plots tended to shift down along Axis 2 (explaining 23% of variation in the distance matrix), consistent with increasing non-native relative abundance, and no clear direction of change was observed for the temporal vectors generated from xeric prairie plots (Fig 2).

## Species that changed the most in each community

*Ventenata dubia* was among the strongest indicator species for the most recent sampling effort in all community types, and the only non-native that increased abundance across our entire study area, particularly within the mesic prairie and old field communities (Table 4). Dominant species compositional changes in the old fields included decreased abundances of two non-native perennial grasses (*Poa compressa* and *P. pratensis*) and native perennial forb species (*Achillea millifolium*, *Lupinus* spp.) and increased abundances of *F. idahoensis*, *V. dubia*

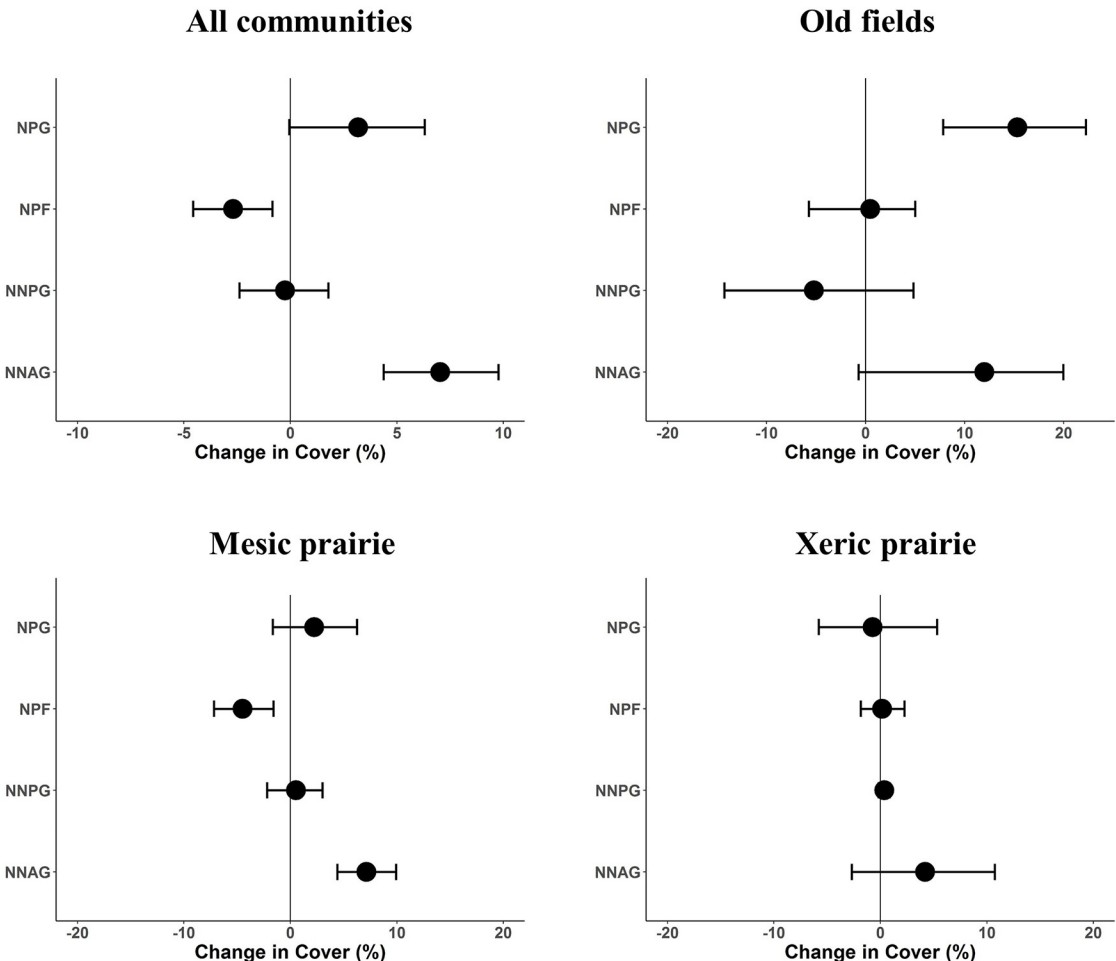

**Fig 1. Foliar cover change (paired differences) between 2008/2009 and 2015/2016 for dominant vegetation functional groups (NPG, native perennial grass; NPF, native perennial forb; NNPG, non-native perennial grass; NNAG, non-native annual grasses) in the Zumwalt Prairie Preserve.** Points are means, error bars contain 95% confidence intervals (bootstrap, 10,000 replicates, percentile method) around the paired difference in the means between the two time periods, and vertical lines indicate zero change.

**Table 3. MRBP comparing vascular plant composition (plots = blocks), between two time periods (2008/2009 and 2015/2016) seven years apart (time period = groups).**

| Vegetation Group 1 –Old Fields | | | | | | | |
|---|---|---|---|---|---|---|---|
| | δ Under null hypothesis | | | | | | |
| Distance Measure | Observed δ | Expected δ | Variance | Skewness | T | A | P |
| Sorensen | 1.48 | 1.60 | 0.36 e-03 | -1.04 | -6.26 | 0.07 | ≪0.001 |
| Vegetation Group 2 –Mesic Prairie | | | | | | | |
| | δ Under null hypothesis | | | | | | |
| Distance Measure | Observed δ | Expected δ | Variance | Skewness | T | A | P |
| Sorensen | 1.25 | 1.32 | 0.82 e-05 | -1.02 | -24.4 | 0.05 | ≪0.001 |
| Vegetation Group 3 –Xeric Prairie | | | | | | | |
| | δ Under null hypothesis | | | | | | |
| Distance Measure | Observed δ | Expected δ | Variance | Skewness | T | A | P |
| Sorensen | 1.14 | 1.18 | 0.46 e-04 | -0.88 | -5.55 | 0.03 | ≪0.001 |

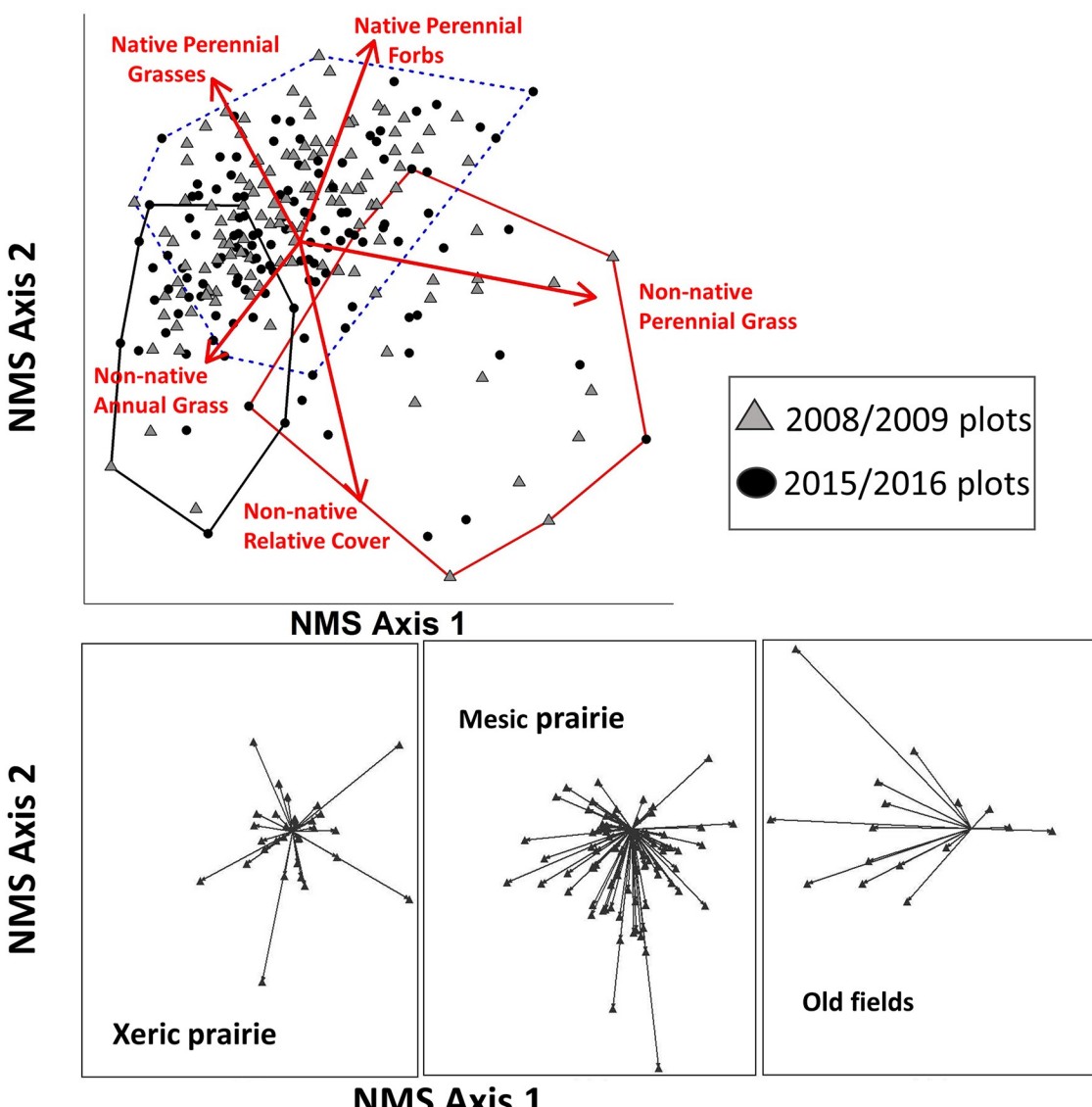

**Fig 2. NMS ordination of plots in species space.** (Top panel) Joint plot with vectors showing direction and magnitude (length) of linear correlations between functional group abundance (% cover) and the ordination space. Convex hulls show the three community types (red solid line, old field community; blue hashed line, mesic prairie community; black solid line, xeric prairie community). (Bottom panels) Temporal vectors showing direction and magnitude of change in ordination space for each plot within three grassland communities. Axis one and two explained 64 and 23% of variation in the distance matrix respectively.

(rank abundance in old fields changed from 19$^{th}$ to 4$^{th}$ over time; [25]), and *Geum triflorum* (Table 4). Decreases in bare ground cover were correlated with increases in cover of *F. idahoensis* (r = -0.12), *V. dubia* (r = -0.41) and *G. triflorum* (r = -0.32). Within the old field community, *V. dubia*, native perennial grass, and native perennial forb increases tended to occur in the same area of ordination space, and were negatively associated with non-native perennial grass cover in old fields (Fig 3).

Dominant species compositional changes in the mesic prairie included substantial increases in non-native species abundance (*V. dubia*, *V. arvensis*, *T. intermedium*) and decreased native perennial forb (*Lupinus spp.*, *A. sororia*, *Potentilla gracilis*, *Hieracium cynoglossoides*, *Geranium viscosissimum*) cover and subtle declines in native bunchgrass (*F. idahoensis*, *P. spicata*)

**Table 4. Indicator species analysis results showing changes in relative cover and frequency (based on proportional abundance of a particular species in a given group relative to that species in all groups; groups being time periods), group assignment and indicator values (IV) for species with the strongest IVs and associated time period tendency for each community.** Time period 1 was 2008 & 2009; time period 2 was 20015 & 2016. Tendency towards time period 1 indicates decreased abundance over time, and tendency towards time period 2 indicates increased abundance over time.

| Species | Rel.Cover δ (%) | Rel.Freq δ (%) | Time period Tendency | IV | p-value |
|---|---|---|---|---|---|
| **Group 1: Old Fields** | | | | | |
| *Poa compressa* | -58 | -43 | 1 | 63 | **0.005** |
| *Achillea millifolium* | -26 | -18 | 1 | 49.9 | 0.148 |
| *Poa pratensis* | -18 | -7 | 1 | 49.9 | 0.125 |
| *Lupinus spp.* | -32 | -13 | 1 | 39.4 | 0.127 |
| *Geum triflorum* | +40 | +12 | 2 | 51.7 | 0.124 |
| *Ventenata dubia* | +70 | +25 | 2 | 57.6 | **0.020** |
| *Festuca idahoensis* | +46 | +6 | 2 | 62.7 | **0.001** |
| **Group 2: Mesic Prairie** | | | | | |
| *Lupinus spp.* | -26 | -13 | 1 | 59.9 | **0.0004** |
| *Festuca idahoensis* | -4 | -1 | 1 | 51.2 | 0.119 |
| *Pseudoroegneria spicata* | -2 | -7 | 1 | 51.1 | 0.142 |
| *Poa secunda* | 0 | -6 | 1 | 49.7 | 0.170 |
| *Arnica sorriea* | -16 | -3 | 1 | 48.8 | 0.186 |
| *Potentilla gracilis* | -22 | -11 | 1 | 48 | **0.003** |
| *Koeleria macrantha* | -12 | -8 | 1 | 47.9 | 0.151 |
| *Hieracium cynoglossoides* | -36 | -22 | 1 | 35.5 | **0.001** |
| *Danthonia intermedium* | -28 | -11 | 1 | 22.9 | **0.065** |
| *Geranium viscossisimum* | -32 | -12 | 1 | 22.2 | **0.023** |
| *Balsomhorriza incana* | -24 | -9 | 1 | 22.1 | 0.113 |
| *Phlox spp.* | -24 | -7 | 1 | 22.1 | 0.119 |
| *Poa compressa* | +32 | +12 | 2 | 17.1 | 0.083 |
| *Thinopyrum intermedium* | +38 | +16 | 2 | 24.0 | **0.014** |
| *Veronica peregrina* | +52 | +38 | 2 | 51.7 | **0.0002** |
| *Ventenata dubia* | +60 | +26 | 2 | 61.0 | **0.0002** |
| **Group 3: Xeric Prairie** | | | | | |
| *Bromus annual spp.* | -40 | -23 | 1 | 63.7 | **0.0002** |
| *Arnica sorriea* | -30 | -25 | 1 | 50.5 | **0.014** |
| *Koeleria macrantha* | -26 | -26 | 1 | 47.1 | 0.062 |
| *Phlox spp.* | -36 | -26 | 1 | 45.5 | **0.038** |
| *Potentilla gracilis* | -24 | -10 | 1 | 31.6 | 0.118 |
| *Danthonia intermedium* | -46 | -22 | 1 | 26.1 | 0.098 |
| *Bromus carinatus* | +62 | +29 | 2 | 33.8 | **0.013** |
| *Poa compressa* | +66 | +23 | 2 | 24.3 | **0.039** |
| *Ventenata dubia* | +10 | 0 | 2 | 52.5 | 0.188 |

abundance (Table 4). *Ventenata dubia* dominance increased substantially (rank abundance in the mesic prairie changed from 11th to 3rd over time; [25]). Native perennial forbs showed a slight tendency to decrease in plots with higher initial non-native annual grass cover as well as where non-native annual grass cover increased over time (Fig 3).

Notable changes in species abundance within the xeric prairie included increased abundances of two non-native grass species (*V. dubia*, *P. compressa*). Other species that showed significant changes in abundance in the xeric prairie were those that tended to be associated with the more mesic communities and were sparse in the xeric prairie community [25], i.e., increased abundance of one native perennial grass (*Bromus carinatus*), and decreased

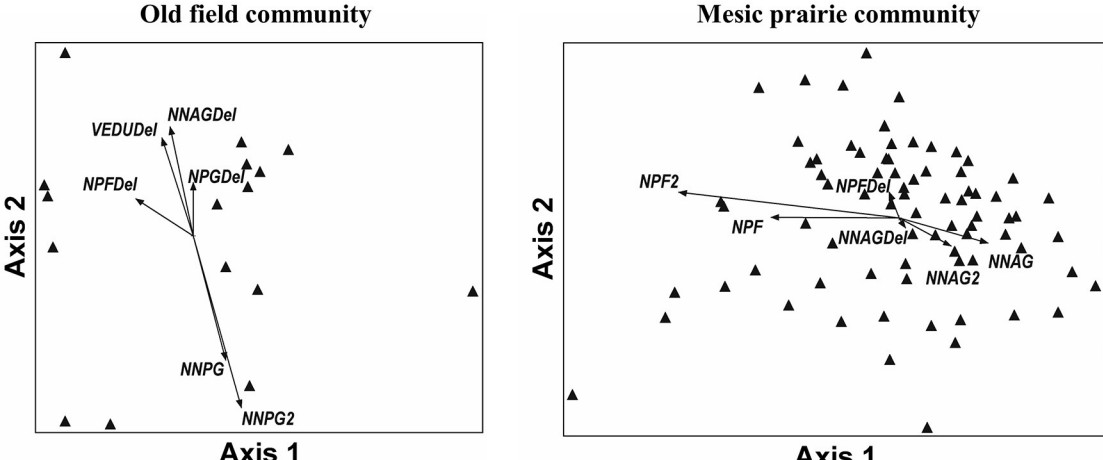

**Fig 3. NMS ordinations of the old field and mesic prairie communities using abundance data from 2015/2016.** The direction and length of the vectors indicate direction and strength of linear correlations between variables and the ordination space. NPG (native perennial grass); NNPG (non-native perennial grass); NPF (native perennial forb); NNAG (non-native annual grass); VEDU (*Ventenata dubia*). "Del" signifies difference between time periods, and a "2" indicates the second time period, e.g., NNPG is the cover of non-native perennial grasses for the 1st time period (2008/2009) and NNPG 2 is cover for non-native perennial grasses for the 2nd time period (2015/2016).

abundances of native perennial forb (*A. sororia*, *Phlox spp.*, *P. gracilis*) and grass species (*K. macrantha*, *D. intermedium*; Table 4).

## Relationship between *V. dubia* and native perennial cover

*Ventenata dubia* cover was negatively related to native perennial cover and native perennial forb cover for both sampling periods (Fig 4 & Table 5). The greatest negative slopes were

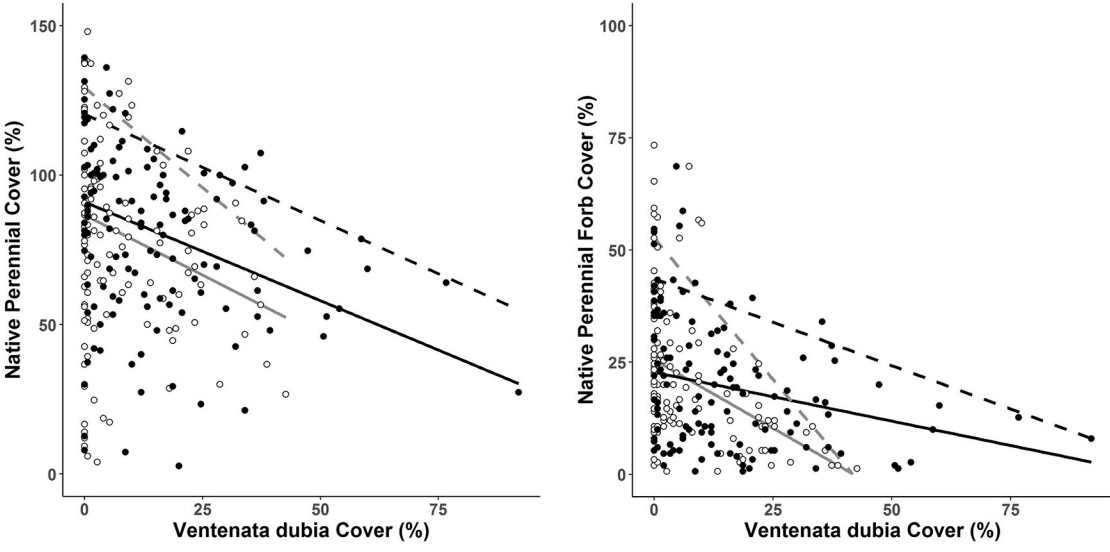

**Fig 4. Scatter plots of *Ventenata dubia* foliar cover against native perennial grass and forb foliar cover (left panel) and native perennial forb foliar cover (right panel).** White points and grey lines indicate data and regression fits from the 2008/2009 sampling period, and black points and black lines show data from the 2015/2016 sampling period. Solid lines are regression lines for the 0.5 quantile and dashed lines are regression lines for the 0.9 quantile.

**Table 5. Quantile regression summary output including response and predictor variables, time, i.e., Time period (1 indicates the 2008/2009 time period; 2 indicates the 2015/2016 time period), slope coefficient and lower and upper bounds of the 95% confidence intervals.** Significant (alpha = 0.05) slope coefficients are indicated in bold.

| Response | Predictor | Time | Quantile | Slope Coeff | Lower bd | Upper bd |
|---|---|---|---|---|---|---|
| Native perennial cover | *V. dubia* cover | **1** | **0.5** | **-0.8** | **-1.21** | **-0.59** |
| | | **1** | **0.9** | **-1.34** | **-1.92** | **-0.59** |
| | | **2** | **0.5** | **-0.67** | **-0.79** | **-0.29** |
| | | **2** | **0.9** | **-0.71** | **-0.86** | **-0.47** |
| Native perennial forb cover | *V. dubia* cover | **1** | **0.5** | **-0.61** | **-0.67** | **-0.52** |
| | | **1** | **0.9** | **-1.26** | **-1.35** | **-0.99** |
| | | **2** | **0.5** | **-0.22** | **-0.5** | **-0.14** |
| | | **2** | **0.9** | **-0.39** | **-0.44** | **-0.25** |
| Native perennial grass cover | *V. dubia* cover | 1 | 0.5 | 0.15 | -0.29 | 0.63 |
| | | **1** | **0.9** | **-0.38** | **-0.63** | **-0.05** |
| | | **2** | **0.5** | **-0.14** | **-0.37** | **-0.12** |
| | | 2 | 0.9 | -0.25 | -0.48 | 0.26 |
| Other Non-native annual grass cover | *V. dubia* cover | **1** | **0.5** | **0.08** | **0.04** | **0.14** |
| | | 1 | 0.9 | 0.18 | -0.01 | 0.70 |
| | | **2** | **0.5** | **0.09** | **0.01** | **0.12** |
| | | 2 | 0.9 | 0.04 | -0.03 | 0.56 |
| Non-native perennial grass cover | *V. dubia* cover | **1** | **0.5** | **-0.18** | **-0.30** | **-0.11** |
| | | **1** | **0.9** | **-1.25** | **-1.44** | **-1.05** |
| | | **2** | **0.5** | **-0.09** | **-0.18** | **-0.05** |
| | | **2** | **0.9** | **-0.05** | **-0.72** | **-0.01** |

detected for the 0.9 quantile followed by the 0.5 quantile (Fig 4 & Table 5). The slopes for responses in both quantiles decreased over time particularly in the 0.9 quantile (Fig 4 & Table 5). Relationships between *V. dubia* cover and native perennial grass cover were inconsistent between years and across quantiles. *V. dubia* cover was negatively related to native perennial grass cover for the 0.9 quantile during the first sampling period but not the 0.5 quantile and for the 0.5 quantile during the second sampling period, but not the 0.9 quantile (Table 5). Because vegetation trends in the old field community were much different than those in the uncultivated prairie, analyses were also performed for the uncultivated prairie alone; however, this did not change our findings or interpretation of the data. Inconsistent relationships between *V. dubia* and native perennial grasses over time and quantiles suggest complex interactions between these two variables and may be related to factors such as lag time effects between species interactions, variation in native species responses, differential responses by community type, or may reflect a weak relationship between these variables that cannot be distinguished from noise in the data. *V. dubia* was negatively related to non-native perennial grass cover during both sampling periods. The greatest negative slope (coefficient = -1.3, 95% CI = -1.4, -1.1) was observed for the 0.9 quantile during the first sampling period. Slopes for both quantiles decreased substantially and were similar for the second sampling period where it was estimated that a one percent increase in *V. dubia* cover corresponded to ~ 0.1% decrease in non-native perennial grass cover for the 0.5 and 0.9 quantiles (Table 5). *V. dubia* was positively related to cover of other non-native annual grasses for the 0.5 quantiles during both sampling periods. It was estimated that a 1% increase in *V. dubia* cover corresponded to approximately a 0.1% increase in the median cover of other non-native annual grasses (Table 5).

## Discussion

Our study revealed three major findings. First, we found evidence of increased relative cover (~ 7%) of native vegetation in old fields, mostly due to increased *F. idahoensis* (increased from an average of 5 to 16%) and *G. triflorum* abundance (increased from 3 to 7%; [25]). Our observations were contrary to a previous study, where native perennial bunchgrass cover did not change consistently over time in ZPP's old fields [17]. This inconsistency may be explained by the low sample size of old field sites (n = 4) analyzed by Taylor and Schmalz (2012) making detection of overall trends difficult among variation across just a few sites. However, our findings are consistent with Taylor and Schmalz's (2012; [17]) hypothesis that *F. idahoensis* abundance would continue to increase in old fields over time. Despite increasing cover of some native species in old fields, two lines of evidence suggest that a continued shift back to native species dominance may not occur: 1) we found no evidence that non-native perennial grass cover has changed over time. In one way this is promising for conservation efforts because it suggests that the non-native perennial grasses have not expanded their distributions. Yet, this finding also suggests that they have not given ground to colonizing vegetation and aligns with previous observations which indicate that non-native perennial forage grasses seeded after disturbance can be highly competitive with colonizing native species and persist on the landscape [36–40]; and 2) *V. dubia* has spread into and quickly become a dominant species in ZPP's old fields [25]. Cultivation of dry land sites, similar to the ZPP, can leave long lasting impacts on site characteristics [17,37,41–42], that can facilitate invasion by non-native species [43] and in some cases may result in the crossing of thresholds which make recovery to pre-disturbance communities unlikely [37,42,44–45]. Cultivation legacies (e.g., increased cover of bareground) may have facilitated the expansion of *V. dubia* into old fields. For example, change in *V. dubia* foliar cover was negatively related (r = -0.41) to change in cover of bare ground in our study. Old fields in the PNB canyon grasslands contain a higher dominance of *V. dubia* in the soil seed bank than similar adjacent uncultivated sites, suggesting that, once introduced, this species is well positioned for continued dominance when native seed is limited [21]. However, our observations that *V. dubia* has continued to spread into as well as become a dominant species in the previously uncultivated mesic prairie suggests that *V. dubia* invasion into the most mesic grassland sites is not dependent on cultivation related disturbance. Therefore, *V. dubia* will likely remain a dominant component in ZPP's old fields (given current conditions, management, and land-use practices) even if native vegetation continues to recolonize that community.

Given our results, we suspect that restoration efforts in ZPP's old fields that reduce non-native perennial grass cover and open up bare ground have the potential for re-colonization by both native vegetation (i.e. *F. idahoensis* and *G. triflorum*) and/or *V. dubia*, and will probably result in colonization by both. Little is known about *V. dubia* impacts on native vegetation abundance and composition or the thresholds/potential to shift dominance from one functional group (native perennial, non-native perennial, or non-native annual) to another in PNB grasslands. We recommend that future research include the identification of potential community states and thresholds for transitions between different communities in PNB grasslands. Within old fields in particular, analyses comparing outcomes for different potential communities may be helpful when considering restoration/management options. For example, these types of analyses may help practitioners address questions such as: (1) would conservation of non-native perennial grass dominated communities limit the spread of *V. dubia*, and provide more desired ecosystem functionality compared to other possible states in old fields?; or, (2) are other potential communities such as a "native perennial grass/*V.dubia*" co-dominant

community possible, and if so, would such a community be more beneficial for conservation of biodiversity?

Our second important finding is that within previously uncultivated mesic prairie, that non-native abundance–primarily *V. dubia*–increased (relative cover increased by ~ 6%; foliar cover increased by a factor of 4) at the same time that native perennial forb abundance declined (from 30 to 25%). Regardless of whether or not these two responses have a causal relationship, both responses–if they are indicative of longer-term trends–may have important consequences to the ecological integrity of the ecosystem. Increased dominance of non-native annual grasses is likely to have substantial impacts on community level functionality including composition, diversity, and soil characteristics of the invaded system [6]. Conversions of bunchgrass dominated communities to non-native annual grasslands in other western North American ecosystems have been associated with reduced floral and faunal diversity, decreased forage production for grazing animals, reduced soil stability, increased susceptibility to erosion, and alteration to disturbance regimes (e.g., increased fire frequency and intensity), nutrient cycling dynamics, and species/structural composition [5,46–51]. Most examples of non-native annual grass impacts on bunchgrass communities in the Intermountain West have focused on annual *Bromus* species (e.g. *Bromus tectorum*; [48,52]). To date, few studies have investigated invasion dynamics and impacts of *V. dubia* [53]. Those that have, suggest potential competitive displacement of native vegetation by *V. dubia* [54, 55], reduced forage production, evidence of a negative association between *V. dubia* and biodiversity in a northern Idaho grassland [53], and a potentially broad ecological niche for *V. dubia* within bunchgrass, shrubland, and open forest communities in the Pacific Northwest [24,53,55–56]. Within Sagebrush-steppe habitats, *V. dubia* cover has been positively associated with mesic upland sites [55] and negatively associated with bio-crust cover and lichen richness [57]. Thus far, research has not identified linkages between fire and *V. dubia* invasion in the PNB [58]. Further research is needed to inform land managers about the potential impacts of *V. dubia* on invaded PNB communities, and whether or not such impacts will mirror those of other annual grass invasions in western North America.

Our detection of a small decline in native perennial forb cover may be an early indication of a longer term trend in native vegetation change; however, we suggest caution in interpretation of the ecological significance of this result for a couple of reasons. First, our study is based on change between two time periods, and a 5% change in foliar cover is a relatively small change that may be attributable to some unmeasured natural variation in the system. Future research will be needed to determine long-term trajectories of perennial forb abundance as well as species-specific responses. Second, we do not expect that an overall ~5% decline in perennial forb cover will have significant impacts to ecosystem function or services in the ZPP. We recommend that trends in native perennial forb abundance be closely monitored at the ZPP because continued declines of forb cover may have negative cascading effects in the ZPP. Native perennial forbs serve important functions in PNB grasslands. Forbs are extremely important for floral diversity in PNB prairies because they are the major source of floral richness (make up > 60% of vascular plant species in the ZPP) in these and similar grasslands [14, 59–60]. Changes in native perennial forb abundance can impact species diversity and abundance at higher trophic levels, i.e., arthropod diversity and grassland birds [61–63]. Pollinator populations are particularly sensitive to changes in forb composition and abundance because forbs increase habitat and forage diversity for pollinators, and their flowers are the primary food source for adult stage pollinators in grasslands [15, 64]. Forbs have functionally important traits in PNB grassland systems including nitrogen fixation by legumes (e.g., *Lupinus* spp) and the presence of taproots which differ in their interaction with soil properties (e.g., water infiltration) compared to fibrous roots of grasses. Reductions in such functional traits can

influence above and below ground floral and faunal composition and diversity, habitat, soil physical characteristics, and nutrient dynamics [65–66]. Additionally, many PNB forbs are culturally important resources (e.g., foods and medicines) for Native American tribes in the region [67–68].

Our third important finding was that *V. dubia* has invaded, and become a dominant species in all three communities (rank abundance = 1, 3, & 4 for the xeric, mesic, and old field communities respectively; [25]), and its abundance was negatively related to native perennial forb abundance. Our finding that *V. dubia* has rapidly spread into the mesic prairie and become a dominant species - 3rd in rank abundance and 6th in rank frequency—absent fire or major anthropogenic disturbances (heavy grazing pressure or known cultivation) suggests that previous hypotheses of high resistance of the more mesic PNB communities to invasion by non-native annual *Bromus* spp. may not apply to *V. dubia* [23,24]. Because this was an observational study, we do not know whether or not *V. dubia* invasion is responsible for the decline in native perennial forbs that we observed. Previous evidence that annual grasses have reduced native and non-native perennial forb abundance through competition for light in similar grasslands [69] coupled with our detection of a negative association between *V. dubia* and native perennial forb cover suggest that a causal link between increased *V. dubia* cover and decreased native forb cover should be explored further. Of course, both of these responses may be consequences of some other factor that is driving change in our study area. Other important disturbances to the ZPP system, including livestock grazing (a novel disturbance compared to evolutionary pressures), wild ungulate herbivory, and fire (a regime greatly altered by human activity), can have profound impacts on vegetation composition and may play important roles in vegetation change that we detected. Most of our study area has been and continues to be grazed by cattle. Livestock grazing has been hypothesized as an important factor that decreased the resilience of many western North American grasslands to annual grass invasion [5,46]. Much of the facilitation of livestock grazing to annual grass invasion is thought to be related to legacy effects of heavy grazing (preceding grazing regulations) that coincided with drought. It is unclear how cattle grazing, today, influences the invasion dynamics of annual grasses in the PNB. Disturbance by high populations of elk (*Cervus canadensis*) may also promote annual grass establishment in the region [70]. A recent study suggests that the combined herbivory by cattle and elk may facilitate *V. dubia* invasion at the ZPP [58]. Additionally, long-term changes in climate may influence changing distributions of species in our study area [71–72]. Future experimental studies that consider the impact of livestock grazing, wild ungulate herbivory, fire, and climate will aid the identification of drivers for *V. dubia* spread as well as the potential for a causal linkage between *V. dubia* invasion and native vegetation change.

## Conclusion

This effort is one of few peer-reviewed studies to evaluate changes in native and non-native plant species composition over time in a PNB grassland (but see [20,58]). Our results demonstrate that plant composition of ZPP's old fields are shifting towards that of the uncultivated grassland plots. Such change in old fields, however, may not represent progress towards native dominated communities because our findings suggest that non-native perennial grass cover has not changed and previous bare ground has not only been colonized by native species but also by *V. dubia* which is now the fourth most dominant species in old fields. Future research is needed to better understand the potential plant community states in PNB old fields, how different community states may aid/hinder management objectives (e.g., conservation of biodiversity), as well as management practices that influence transitions between potential community states. While we found varying responses of native vegetation across community

types, in general, our findings indicate that despite the protected status of the ZPP, that *V. dubia* (an annual non-native grass) has rapidly spread into and become a dominant species within all major grassland community types (xeric, mesic, and old fields) in the ZPP. Previous research in similar grasslands have consistently demonstrated that shifts from native bunch-grass to non-native annual grass dominance greatly alter ecosystem functionality and are generally associated with suppressed native species richness and abundance. Such knowledge combined with our findings that *V. dubia* cover was negatively related to native perennial forb cover, as well as our detection of potential declines in native perennial forb abundance in the most recently invaded grassland sites suggest that *V. dubia* poses a threat to conservation of biodiversity and grassland function in the largest remaining protected Pacific Northwest Bunchgrass remnant.

## Acknowledgments

We thank Kent Coe, Andie Lueders, Tyler Dungannon, Kolby Haliewicz, Walker Powell, JoAnna Perren, Luke Ridder, Brogan Watson, Ashtin Olsen, Ngeyaol Polycarp, and Shanda Zettles for assistance collecting field data; The Nature Conservancy for their collaboration and support for this project; and anonymous reviewers for suggestions on the manuscript. Funding was provided by USDA NRI Competitive Grant Agreement No. 2006-35320-17244 (BE), USDA Forest Service Pacific Northwest Research Station # 05-JV-11261967-069 (BE), the Oregon State University Agricultural Research Foundation (LM, BE), and the Oregon State University Agriculture Experiment Station (LM, BE).

## Author Contributions

**Conceptualization:** Joshua P. Averett, Lesley R. Morris, Bridgett J. Naylor, Robert V. Taylor, Bryan A. Endress.

**Data curation:** Joshua P. Averett, Bridgett J. Naylor, Robert V. Taylor.

**Formal analysis:** Joshua P. Averett.

**Funding acquisition:** Lesley R. Morris, Bryan A. Endress.

**Investigation:** Joshua P. Averett, Bridgett J. Naylor, Robert V. Taylor, Bryan A. Endress.

**Methodology:** Joshua P. Averett, Bridgett J. Naylor, Bryan A. Endress.

**Resources:** Bridgett J. Naylor, Robert V. Taylor.

**Supervision:** Lesley R. Morris, Bryan A. Endress.

**Writing – original draft:** Joshua P. Averett, Lesley R. Morris, Bridgett J. Naylor, Robert V. Taylor, Bryan A. Endress.

**Writing – review & editing:** Joshua P. Averett, Lesley R. Morris, Bridgett J. Naylor, Robert V. Taylor, Bryan A. Endress.

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
