## [Decision Letter · Decision Letter 0]

27 Oct 2019

PONE-D-19-27436

Vegetation change over seven years in the largest protected Pacific Northwest Bunchgrass Prairie remnant.

PLOS ONE

Dear Mr. Averett,

Thank you for submitting your manuscript to PLOS ONE. After careful consideration, we feel that it has merit but does not fully meet PLOS ONE’s publication criteria as it currently stands. Therefore, we invite you to submit a revised version of the manuscript that addresses the points raised during the review process.

I would like to see all of Reviewer 1's comments addressed before re-submission. You need to provide detail regarding any treatments that may have occurred in the time interval between the two sampling periods, and address how those treatments likely influenced your results. The reviewers both felt the manuscript was well-written, and provided information that will be a useful contribution to the literature on the topic. After reading your manuscript myself, I had many of the same questions/comments raised by Reviewer 1, and that was the basis of my decision.

We would appreciate receiving your revised manuscript by Dec 11 2019 11:59PM. To enhance the reproducibility of your results, we recommend that if applicable you deposit your laboratory protocols in protocols.io, where a protocol can be assigned its own identifier (DOI) such that it can be cited independently in the future. For instructions see: http://journals.plos.org/plosone/s/submission-guidelines#loc-laboratory-protocols

We look forward to receiving your revised manuscript.

Kind regards,

Matt A Bahm, Ph.D.

Academic Editor

PLOS ONE

Journal Requirements:

Additional Editor Comments:

Overall, this is a well-written manuscript that provides interesting/useful information. Reviewer 1 points out several major issues that need to be addressed in your manuscript, and I concur with their recommendations. Generally, you need to address the influence of treatments during the time interval of your study that could have influenced your results.

Reviewers' comments:

Reviewer's Responses to Questions

**Comments to the Author**

1. Is the manuscript technically sound, and do the data support the conclusions?

Reviewer #1: Yes

Reviewer #2: Yes

2. Has the statistical analysis been performed appropriately and rigorously? 

Reviewer #1: Yes

Reviewer #2: Yes

3. Have the authors made all data underlying the findings in their manuscript fully available?

Reviewer #1: Yes

Reviewer #2: Yes

4. Is the manuscript presented in an intelligible fashion and written in standard English?

Reviewer #1: Yes

Reviewer #2: Yes

5. Review Comments to the Author

Reviewer #1: Review or PONE-D-19-27436

This manuscript summarizes changes that occurred in a PNW bunchgrass prairie community over time. While this is important to document, particularly with increasing threats to this sensitive ecosystem, the paper is lacking valuable details (besides a brief mention of Rx fire, grazing and herbicide in the Methods) about what management has occurred across this site over time and how those actions may have influenced the shifting communities. There is substantial literature of the effects of each of these management tools on native and non-native plant communities and a more thorough discussion of these impacts would strengthen the paper and provide some potential mechanisms driving the findings. I have therefore recommended 'major revision', as this could add a substantial component to the methods, results & discussion, especially is there is the opportunity to group results based on management histories (that will depend on if/how well management actions have been documented for the sampled areas).

Methods:

It would be helpful to know what restoration/management treatments (Rx fire, targeted herbicide, native seeding/planting, etc.) were applied within each of these plant community types over the 7 years between sampling times that could have influenced the resulting plant communities. For instance, what is the fire frequency? What grazing rotations are used? What herbicides are used and how are they applied? Have any native plants been (re)introduced into treated areas?

Ln194: How were the plots distributed among the different plant community types?

Ln202: Change ‘were’ to ‘was’

LN232: Remove comma after ‘performed’

Ln240-242: This sentence has awkward punctuation- I’d recommend separating into two sentences.

Results:

Ln273: Remove the comma after V. dubia

LN314: You did not include measurement of bare ground in your Methods.

Ln322: Why is the decline in perennial bunchgrasses described as ‘potentially subtle’ (vs. just ‘subtle’)?

Discussion:

Ln363: Why do you think your findings disagreed with the previous study?

Ln383-389: Run-on sentence – consider splitting into two sentences

Ln400: number these questions

Ln402: remove comma after ‘states’

Ln404: the ideas presented here seem like a good opportunity to incorporate some examples and ideas surrounding the concept of novel ecosystems.

Ln443: trophic is misspelled

Ln447: Not sure ‘novel’ is the appropriate word to use here, as that suggests that these functional traits are new. Maybe ‘functionally important’ instead?

Ln454: I’d recommend starting this paragraph with ‘Our third important finding’ instead of ‘Thirdly’ to be consistent with previous paragraphs.

Ln459: add ‘s’ to ‘suggest’

Ln468-470: add comma after ‘system’ and ‘…human activity)’

Ln472-475: This would be a good place to discuss the restoration and/or management actions that have been applied in these sites over the past seven years and how they may have influenced your findings.

Reviewer #2: This is a nice summary of vegetation change over time and it uses clear and appropriate statistical methodology to look at both individual species and whole community change. You clearly address the limitations of an observational study and point out important directions for future experimental research. Your interpretations of the trends that stand out in your analyses are concise and well-supported by external evidence and you do a good job of discussing the potential for many other unmeasured drivers. This is an important initial look at change in this threatened ecosystem and will be helpful to guide future research in the Zumwalt Prairie and in other remnant PNW prairie ecosystems. I've attached a pdf with just a few additional suggestions on the figures.

6. PLOS authors have the option to publish the peer review history of their article (what does this mean?). If published, this will include your full peer review and any attached files.

Reviewer #1: No

Reviewer #2: No

---

## [Author Response · Author response to Decision Letter 0]

11 Dec 2019

Dear Editor:

We are pleased to resubmit our revised manuscript: “Vegetation change over seven years in the largest protected Pacific Northwest Bunchgrass Prairie remnant”; Manuscript no. PONE-D-19-27436. I would like to thank you and the reviewers for the helpful comments and suggestions to improve the quality of our manuscript. Please see our responses to each specific comment in italicized blue text. Thank you and please let me know if there are any additional edits needed. We are excited to move forward with this manuscript in PLOS ONE.

Associate Editors Comments: 

Thank you for submitting your manuscript to PLOS ONE. After careful consideration, we feel that it has merit but does not fully meet PLOS ONE’s publication criteria as it currently stands. Therefore, we invite you to submit a revised version of the manuscript that addresses the points raised during the review process.

I would like to see all of Reviewer 1's comments addressed before re-submission. You need to provide detail regarding any treatments that may have occurred in the time interval between the two sampling periods, and address how those treatments likely influenced your results. The reviewers both felt the manuscript was well-written, and provided information that will be a useful contribution to the literature on the topic. After reading your manuscript myself, I had many of the same questions/comments raised by Reviewer 1, and that was the basis of my decision.

Reviewer 1 Comments:

This manuscript summarizes changes that occurred in a PNW bunchgrass prairie community over time. While this is important to document, particularly with increasing threats to this sensitive ecosystem, the paper is lacking valuable details (besides a brief mention of Rx fire, grazing and herbicide in the Methods) about what management has occurred across this site over time and how those actions may have influenced the shifting communities. There is substantial literature of the effects of each of these management tools on native and non-native plant communities and a more thorough discussion of these impacts would strengthen the paper and provide some potential mechanisms driving the findings. I have therefore recommended 'major revision', as this could add a substantial component to the methods, results & discussion, especially is there is the opportunity to group results based on management histories (that will depend on if/how well management actions have been documented for the sampled areas).

Methods:

It would be helpful to know what restoration/management treatments (Rx fire, targeted herbicide, native seeding/planting, etc.) were applied within each of these plant community types over the 7 years between sampling times that could have influenced the resulting plant communities. For instance, what is the fire frequency? What grazing rotations are used? What herbicides are used and how are they applied? Have any native plants been (re)introduced into treated areas?

We added text in a couple of places in the manuscript:

(1) In lines 188-200; we indicate that cattle were grazed over most of our study area (~ 95% of plots) at stocking rate of 4.7 acres per AUM consistent with the larger Zumwalt Prairie and the last several decades of grazing management in the area. We also indicate that 12 plots (~ 10%) were exposed to some level of management/restoration activities (e.g., prescribed fire, n = 4; recent or temporary exclusion of cattle grazing, n = 6, herbicide and seeding, n = 1) over the duration of our study [25]. We have updated our data article (which makes available all of the data used in this study) to include any management/restoration treatments by plot so that this information is transparent to readers. The data article and raw data files have been accepted with revisions to Data In Brief The small sample size of plots with any specific management treatment did not allow for the effects of such treatments to be evaluated in this study. Additionally, we acknowledge that we ran all analyses with and without the 12 plots, and there inclusion/deletion did not alter our findings, and did not change the significance (at an alpha level of 0.05) of any comparisons or relationships in our analyses. Therefore, we decided to include all of the plots in the analyses to capture more variation in vegetation composition across the prairie habitat of the Zumwalt Prairie Preserve.

(2) We clarify (in lines 134-136) that any herbicide spraying on the Zumwalt Prairie Preserve consists of spot-spraying of individual plants of primarily Hieracium caespitosum, Potentilla recta, and Onopordum acanthium; these sites are currently localized and not widespread. Moreover, none of these common in our study area (not in Table 1).

(3) We added text in lines 492-500 to acknowledge that cattle grazing may influence patterns of invasion that were detected since livestock grazing has been hypothesized to decrease the resilience of western North American grasslands to annual grass invasion. We also provide evidence from the literature where one study found a relationship between elk disturbance and annual grass establishment, and another example that found evidence for the combined effect of cattle and elk herbivory on facilitation of V. dubia invasion at the Zumwalt Prairie Preserve.

Ln194: How were the plots distributed among the different plant community types?

(Now line 213): We added sample sizes (parenthetically) for each of the community types to show the representation of each community in sampled plots.

Ln202: Change ‘were’ to ‘was’

Done.

LN232: Remove comma after ‘performed’

Done. 

Ln240-242: This sentence has awkward punctuation- I’d recommend separating into two sentences.

Done.

Results:

Ln273: Remove the comma after V. dubia

Done.

LN314: You did not include measurement of bare ground in your Methods.

We added text, in lines 208-212, to identify soil characteristics measured (litter, rock, moss/lichen, bare ground) as well as a description of how bare ground cover was calculated in our study: “Plot cover of bare ground was calculated as the percentage of hits (out of 150) for bare ground in that plot where no other plant species or soil cover (litter, rock, moss/lichen) was encountered.”

Ln322: Why is the decline in perennial bunchgrasses described as ‘potentially subtle’ (vs. just ‘subtle’)?

Deleted “potentially”.

Discussion:

Ln363: Why do you think your findings disagreed with the previous study?

We added text (Ln 383-386) indicating that the inconsistency between our finding and Taylor and Schmalz (2012) may have been explained by the low sample size (n = 4) of old field sites analyzed in the previous study which would likely have made detection of overall trends in the data difficult among variation of just a few sites.

Ln383-389: Run-on sentence – consider splitting into two sentences

(Now Ln404-409) We split this sentence into two sentences, and re-worded for clarity.

 Ln400: number these questions

Done.

Ln402: remove comma after ‘states’

Done.

Ln404: the ideas presented here seem like a good opportunity to incorporate some examples and ideas surrounding the concept of novel ecosystems.

We agree, however, this paper does not focus on the future question suggested here. Technically, the entire study area may fit into a novel ecosystem concept (i.e., non-native species are dominant across the prairie), but for the sake of not greatly expanding the manuscript beyond the scope of the research objectives we chose to not expand into the novel ecosystems literature here.

Ln443: trophic is misspelled

Corrected spelling error.

Ln447: Not sure ‘novel’ is the appropriate word to use here, as that suggests that these functional traits are new. Maybe ‘functionally important’ instead?

Changed to “functionally important”.

Ln454: I’d recommend starting this paragraph with ‘Our third important finding’ instead of ‘Thirdly’ to be consistent with previous paragraphs.

(Now line 466): Changed to “Our third important finding”.

Ln459: add ‘s’ to ‘suggest’

Done.

Ln468-470: add comma after ‘system’ and ‘…human activity)’

Done.

Ln472-475: This would be a good place to discuss the restoration and/or management actions that have been applied in these sites over the past seven years and how they may have influenced your findings.

(Lines 188-200 & 492-504): We added a more detailed description of land use treatments and/or lack thereof in the sites sampled. Most of the sites were grazed by cattle at a stocking rate of 4.7 acres per animal unit month (AUM). This livestock grazing system is consistent with stocking rates across the larger Zumwalt Prairie and have been relatively constant for at least 50 years in the area. Twelve plots were subjected to management/restoration treatments; these treatment levels by plot are now included in the data paper (in review with Data In Brief) which will provide the raw data used in this study to readers. Deletion of all 12 plots subjected to management/restoration treatments did not alter our findings so they were retained in the dataset so that we could capture more variation of the community in our dataset.

Reviewer 2 Comments (summarized):

Figures:

Figure 2 – Remove ordination axis labels

Done.

Figure 2 – Ordination colors need to be more distinguishable; distinction of black and blue is difficult

We made the convex hull lines thicker and changed the blue line type to a hashed line so that the blue and black lines (and corresponding community groups) are more easily distinguishable.

Figure 3 – Recommend converting blue symbols to a black and white scale.

We converted the blue symbols to black in the ordinations for Figure 3.

Sincerely,

Josh Averett

Faculty Research Assistant

Eastern Oregon Agriculture and Natural Resource Program

Oregon State University

Badgley Hall, 205

541-962-3249

---

## [Editor Report · Decision Letter 1]

18 Dec 2019

Vegetation change over seven years in the largest protected Pacific Northwest Bunchgrass Prairie remnant.

PONE-D-19-27436R1

Dear Dr. Averett,

We are pleased to inform you that your manuscript has been judged scientifically suitable for publication and will be formally accepted for publication once it complies with all outstanding technical requirements.

With kind regards,

Matt A Bahm, Ph.D.

Academic Editor

PLOS ONE
---

## [Editor Report · Acceptance letter]

16 Jan 2020

PONE-D-19-27436R1 

Vegetation change over seven years in the largest protected Pacific Northwest Bunchgrass Prairie remnant. 

Dear Dr. Averett:

I am pleased to inform you that your manuscript has been deemed suitable for publication in PLOS ONE. Congratulations! Your manuscript is now with our production department. 

With kind regards,

on behalf of

Dr. Matt A Bahm 

Academic Editor

PLOS ONE